# The alignment model of indirect communication

**Asya Achimova**[1]*, **Michael Franke**[1,2], **Martin V. Butz**[2,3]

**1** Department of Linguistics, University of Tübingen, Tübingen, Germany, **2** Department of Computer Science, University of Tübingen, Tübingen, Germany, **3** Department of Psychology, University of Tübingen, Tübingen, Germany

* asya.achimova@uni-tuebingen.de

**Data availability statement:** All the data, data analysis, and modeling files are available from the following OSF repository https://osf.io/khr2j/?view_only= 86a0546483354ef49ad37c58e2cb4f0f

## Abstract

Speakers often choose utterances under uncertainty about the potential opinion of the listener. In this case, utterances that do not signal the speaker's opinion directly may allow the speaker to avoid possible conflict: saying that an election outcome is interesting rather than amazing, even if the speaker is truly excited about it, may give her an option to retreat if it turns out that the listener's opinion is the opposite. By enhancing the Rational Speech Act framework with a turn-taking pragmatic system, we develop a model of indirect communication that is able to (1) rationalize the choice of indirect utterances when speakers' opinions do not align; (2) capture complex reasoning about the true interlocutor's opinion when facing indirect utterances and responses. The model has several novel features: in addition to standard informativeness goals, speaker choices factor in potential divergences of opinions between conversation partners. The listener model further considers multi-turn dialogues rather than isolated utterances: it is able to derive that an utterance like "interesting" can be interpreted positively or negatively depending on preceding discourse. The model, though complex, makes novel, non-trivial qualitative predictions, which are supported by data from three behavioral experiments reported here.

## Introduction

Human linguistic communication is not a simple encoding-decoding process based on a fixed code, but relies heavily on pragmatic reasoning about the context and the conversation partners' beliefs and intents [1]. Consider how the meaning of the predicate "interesting" changes in contexts (1) and (2):

(1) a. Alex: The election outcome is amazing!

 b. Bo: I find it interesting.

(2) a. Alex: The election outcome is awfull!

 b. Bo: I find it interesting.

In (1) Bo's utterance likely signals a negative evaluation, while the same utterance carries a positive evaluation in (2). In this paper we aim to (i) empirically verify the intuition that

**Funding:** AA received funding from the the Deutsche Forschungsgemeinschaft (DFG, German Research Foundation) – Project number 519109539. The work of MB is funded by the Deutsche Forschungsgemeinschaft (DFG, German Research Foundation) under Germany's Excellence Strategy—EXC number 2064/1—Project number 390727645. https://www.dfg.de/. The funding agency did not play a role in study design, data collection and analysis, decision to publish, or preparation of the manuscript.

**Competing interests:** The authors have declared that no competing interests exist.

the meaning of utterances changes dynamically depending on the preceding context; and (ii) develop a formal model of pragmatic inference that is able to reproduce this effect. To develop this model, we first ground the meaning of predicates, such as "amazing" and "interesting" in relation to a one-dimensional scale reflecting an agent's attitude towards a binary issue, and represent an agent's opinion state as a distribution over such scales. Second, we develop a speaker model that regulates utterance choice taking into account the opinion of the listener. Finally, we use Bayesian inference to model how true beliefs of the speaker can be inferred from her utterance and preceding discourse.

Building on the seminal work of Paul Grice [2] and fueled by a more recent "empirical turn" [3,4], formal approaches to pragmatics, and particularly the Rational Speech Act framework [5], have spawned probabilistic models of contextualized reasoning and expression choices, which account for many different phenomena, including the meaning of number words [6], scalar implicature [7], metaphor [8], reference [5,9], and vagueness [10,11] to name but a few cases (for a recent overview see [12]).

The prevalent focus of formal, probabilistic, and experimental pragmatics has long been on a particular mode of language use, namely the cooperative communication of relevant information about the world. A few exceptions include models of strategic communication where values of conversation partners do not align [13,14] and models of use-conditional meaning [15]. More recently, the field is undergoing a second, this time "social turn," focusing increasingly on more social aspects of meaning and communication, including models of signaling one's persona [16,17] and ideology [18,19], learning about others [20,21], and appearing polite [22,23]. Recent experimental studies have targeted the role of face management in determining the meaning of scalar expressions [24]. Along related lines, the interactions of power, social distance, and gender as well as their influence on face management strategies in verbal communication were studied [25].

This work seeks to contribute to this growing literature on probabilistic modeling and experimental investigation of social factors in language use. We introduce a novel extension of the Rational Speech Act family of models, which considers the gradual, dynamic, multi-turn sharing of opinions through the strategic use of indirect messages and pragmatic reasoning in cases where conversationalists may have reason not to reveal their beliefs, stances, and opinions directly.

## Background

Sharing mental attitudes, such as beliefs, preferences, and assumptions—which we here collectively address as *opinions*—is a critical component of interpersonal relations, group formation, and bonding [26,27]. The motivation to share mental states with other people develops early in life. It manifests itself already in the apparent desire of infants to share significant experiences with their caretaker before their first birthday [28]. Experimental evidence further suggests that preschoolers prefer puppet toys that are similar to themselves in physical appearance and food preferences [29]. The "like me/not like me" dichotomy is important already to pre-linguistic infants, who prefer others who share similar traits with them [30]. Mahajan and Winn [30] further maintain that similarity to self is an inherent preference exhibited by humans and further emphasize the importance of similarity for interpersonal attraction. Dissimilarity and conflicts in beliefs and attitudes, in turn, may damage the relationship between interacting partners. During a conversation, monitoring whether an utterance carries a risk to the relationship is one of the factors that determines the speaker's utterance choices. For example, Brown and Levinson [31] conceptualized such social considerations in the notion of *face* and argued that face preservation is a major motivational

force that shapes human interactions. As a result, in social interaction contexts humans are confronted with the objectives to align with their conversation partners while staying true to themselves.

Aligning opinions requires care, restraint, and decency. Consider the case of two researchers in Cognitive Science, Alex and Bo, seeing a poster for a big conference on Machine Learning. If Alex believes that the poster offers promising tools for cognitive science, she may choose a statement, such as in (3).

(3)   These results carry an incredible potential for Cognitive Science!

However, if she fears that Bo's opinion might be negative, a safer option would be an utterance, such as in (4).

(4)   These results carry an interesting potential for Cognitive Science!

The predicate "interesting" in (4) is an example of the kind of indirect language use we focus on in this paper: indirect utterances do not fully reveal the true opinion of the speaker. Additionally, indirect utterances are compatible with multiple interpretations, just like ambiguous images can be interpreted in multiple ways depending on a viewpoint. What changes the interpretation of such utterances are the assumptions of the listener about what opinions are plausible. From Bo's reaction to the utterance in (4), as exemplified in (5), Alex may be able to infer a great deal about Bo's beliefs about the matter.

(5)   a.   Yes, we will soon all be unemployed.
           [negative consequence]

      b.   Yes, we will see exciting new work in computational cognitive modeling.
           [positive consequence]

If Bo replies with the utterance in (5a), then most likely she perceived "interesting" as a negative assessment, while the utterance in (5b) signals her positive perception of the same utterance. The model developed in this paper, the *Alignment Model of Indirect Communication* (AMIC), intends to capture exactly this kind of use of indirect language to explore whether or where opinions—used here as a catch-all term for various stances and attitudes—are shared, so as not to risk loss of face.

In linguistics and philosophy of language, the topic of indirect communication has traditionally been associated with the Speech Act Theory [32,33]. In this framework, speech acts are classified as indirect when they feature a mismatch between their form and function: for example, the utterance in (6) contains a request despite bearing the form of a question.

(6)   Could you please pass me the salt?

While the speech-act-theoretic analysis viewed the indirect/direct distinction as a binary contrast, more recent experimental studies suggest that it might be better viewed as a gradable category (see [34] for an overview of different approaches).

Much of both theoretical and experimental literature on indirect communication has been concerned with formulating the strategies that explain the choice of indirect speech acts, with face-saving being one of the prominent driving forces of indirectness [35]. By choosing to be indirect, the speaker can avoid sending negative feedback to the listener and thus preserve the listener's face. Furthermore saying positive things might be beneficial for the speaker's face as well, since social norms might favor being positive—the so-called Pollyanna effect [25,36–38]. Thus, face-saving strategies have been used as explanations of polite language use [39].

AMIC builds on related earlier work on probabilistic models of politeness (e.g. [22,23,40] and other models of social meaning (e.g.[16,17,41]). Such models generalize more basic models of pragmatic language generation, which assume that the speaker's preferences for selecting utterances pivot almost solely around considerations of truth and informativity (about the world). To explain social language use, these models assume that the speaker's preferences also include a social payoff component, next to informativity. The idea that speakers combine social and informational goals in the choice of their utterances has been previously formulated in both conversation analysis [42] and theoretical pragmatics, which particularly emphasizes the co-presence of informational and relationship goals in conversations [31,43]. The study of language as a social action [44] further emphasizes the role of social considerations in shaping conversations.

A key observation of previous modeling is that, if we define the speaker's preferences for utterances as a composite utility function roughly like

$$\text{Utility}(u) = \omega\,\text{Informativity}(u) + (1 - \omega)\,\text{SocialValue}(u),$$

then we find that speakers tend to produce less informative utterances—here addressed as *indirect* [14,33,45,46]—to the extent that informativity and social value of an utterance diverge and the speaker prefers to emphasize the social dimension of language use over the informative, as captured by the weight parameter $\omega$. The parameter $\omega$ can take values between 0 and 1. A value of 1 means that only informativity will be taken into account; on the contrary, a value of 0 means that no weight is given to informativity but only social value matters, which is often defined as sending positive feedback to the listener in politeness models; a value of $\omega = 0.5$ will distribute the utility evenly between informativity and social goals. For example, a speaker who finds listener's baking skills relatively poor may prefer a double negative "not bad", as in (7) if she weighs social utility highly (low omega). Indirectness here offers the speaker a good evasive strategy if one wants to be polite (for an overview of the effect of negation on the meaning of adjectives, see [25,47]).

(7)  a.  Alex: How did you like my cookies?

  b.  Bo: They were not bad.

Indirect language use is associated with ambiguity—a situation where an utterance features multiple possible meanings. More specifically, we will be concerned with pragmatic ambiguity—a property of utterances that emerges in discourse when the meaning of the utterance as a whole may change depending on the context [48,49], world knowledge, or beliefs of conversation partners. Evidence from computational modeling suggests that ambiguity may offer necessary flexibility to conversation partners to adjust word meanings to each other [50].

The linguistic source of pragmatic ambiguity in our case lies in subjective predicates and their potential to communicate speaker's opinion indirectly. The meaning of subjective predicates, such as "delicious" and "horrible" differs from objective predicates, such as "square" and "deciduous," and these differences affect the types of linguistic environments in which these predicates can occur [51–55]. Critically for us, subjective predicates are gradable—a property that has been related to vagueness [56]. Vagueness of predicates, as a lexical property, can become a source of ambiguity at the utterance level. The usage of partially gradable subjective predicates allows us to construct utterances whose interpretation is influenced by linguistic and extra-linguistic context. Furthermore, subjective predicates allow for so-called faultless disagreement [39,57,58]. In such situations, speakers who express opposite opinions, such as in (8), may still both be right.

(8) Alex: The election outcome is amazing!
 Bo: The election outcome is not amazing!

In politeness contexts, speakers were shown to prefer negated indirect statements such as "not amazing" since they offer speakers several benefits. First, a speaker can avoid using an alternative negative statement, such as "terrible" (see [59] and [60] for an overview of the role of alternatives in speech production and interpretation). Second, negated indirect statements are compatible with a wide range of world states, offering speakers a loophole if they want to deny having given a negative assessment [39].

## Contribution

So far, most probabilistic models of social or polite language use have focused on situations where the social value function was essentially known to the speaker. This is a reasonable assumption, for example, for the case of evaluating the impact of statements like (7) on the addressee's self-esteem. But to explain the case of dynamically exploring potential (mis-) alignment of opinion like in (5), AMIC will include a novel social value function, which is (i) subject to uncertainty, and (ii) considers utility arising from alignment of opinions, with (higher-order) uncertainty about these opinions.

The second and arguably more critical contribution of this work is the inclusion of *multi-turn inferences*, which may capture the dynamic process of opinion alignment as the dialogue unfolds over several turns. Previous attempts to model multi-turn interactions have concerned question-answer pairs [61], dialogues that aim at establishing references to location [62], and sequential language games where agents cooperate to jointly establish the identity of references given that each of the agents only has partial information about the target object [63]. In this work, we turn to the inferences conversation partners draw about each other rather than objects in the world by reasoning about sequences of utterances that have been added to the conversation previously. This inference process allows conversation partners to establish the meaning of utterances in a dialogue.

While AMIC is generally able to model multi-turn inferences across many turns of dialogue, we here focus on two-turn dialogues for reasons of complexity. Just as established work in Conversation Analysis has shown that much can be learned from studying pairs of dialogical utterances, so-called adjacency pairs [64,65], we argue that our focus on two adjacent turns in a conversation, rather than isolated utterances, on the one hand, or unstructured dialogue, on the other hand, enables systematic investigation of basic mechanisms of opinion inference during conversations.

Even with the restriction to two-turn conversations, AMIC showcases how (i) speakers may strategically choose to be indirect and (ii) how they learn about their partner's opinion from the subsequent (linguistic) reaction to their (in)direct utterance. Indeed, we present novel empirical data that supports AMIC's general predictions about the effects on speaker choices of utterances arising from beliefs about mutual opinion and speaker's weighing of social value. More strikingly, based on simulation studies, we find that the model makes novel, subtle, but non-trivial predictions about interpretations in two-turn dialogues, which we test empirically as well.

The paper is structured as follows. We first introduce the vanilla RSA architecture and discuss previous RSA-extensions that introduce complex utility functions. We then propose a novel model which implements social utility from opinion alignment. We showcase interesting predictions of this model, derived from simulation. Finally, we demonstrate that the qualitative patterns predicted by the model are confirmed with behavioral data.

## The alignment model of indirect communication

The goal of the Alignment Model of Indirect Communication (AMIC) is to account for (i) the choice of indirect utterances when the speaker pursues multiple conversational objectives; (ii) the inferences the speaker draws about the opinion of the listener upon observing the listener's reply to an indirect utterance. We develop the AMIC within the Rational Speech Act framework [5,9,66], which models pragmatic communication in terms of speakers choosing utterances to maximize their conversational goals, and listeners using inverse reasoning about the speakers' policy to infer the speakers' intended messages from the observed utterance. Building on previous models in this tradition, we introduce a social utility function to model the speaker's goal of avoiding apparent conflict in opinion and consider a larger interpretation horizon of multiple utterance. To do so, we introduce a novel formalization of (higher-order) belief about opinions, and explore several measures of opinion alignment in the context of the model. The following first introduces a vanilla formulation of the RSA model and previous extensions of it to cover social reasoning, before introducing the AMIC.

### Vanilla RSA

The vanilla RSA model defines probabilistic choice policies for the speaker and the listener. The model contains a specification of the utterance space (a set of utterances that are available to the speaker) and the meaning space (a set of world states that are considered). The speaker selects an utterance $u$ for a given state (a meaning to be communicated) with conditional probability $P_{S_1}(u \mid s)$, which is proportional to the utterance's utility $U_{S_1}(u,s))$ for state $s$, using a parameterized soft-max function:

$$P_{S_1}(u \mid s) \propto \exp(\alpha \cdot U_{S_1}(u,s)) \qquad (1)$$

For example, the scenario in Fig 1 consists of three objects that represent the considered world states and four possible utterances that correspond to the features of the objects: *cloud, circle, blue*, and *red*. The speaker then considers for each of the utterances how likely they are to let the listener identify the intended object. For example, if the speaker wants to signal the object in the middle, she can use the utterances *cloud* or *red*. Both of the utterances are in this case ambiguous since they select two objects, so the speaker assigns equal probability to both of them.

For simple applications, e.g., for referring to an object from a list of potential referents or to describe a world state from a known set of alternatives, the utility function of the speaker can be defined in terms of the negative surprise of a literal interpreter $L_0$:

$$U_{S_1}(u,s) = \log P_{L_0}(s \mid u) \qquad (2)$$

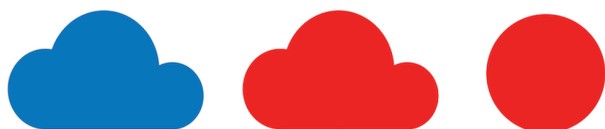

**Fig 1. Scenario: a blue cloud, a red cloud, and a red circle**. If the speaker $S_1$ wants to refer to the object in the middle she can say either *red* or *cloud*, since these utterances give an equal chance that the listener will select the red cloud.

This utility function can alternatively be motivated as the goal of minimizing the distance between the speaker's belief (which is here assumed to be a degenerate probability distribution ruling out all but one world state $s$) and a literal interpreter's belief after hearing utterance $u$ [10,67,68].

The literal listener can essentially be thought of as a construct to ground out pragmatic reasoning in a base layer of semantic meaning, and may simply be defined as a choice of a state proportional to how true it is, for some (Boolean or non-Boolean) semantic meaning function $f$ that maps pairs of utterances and states onto numbers in the unit interval:

$$P_{L_0}(s \mid u) \propto \mathtt{f}(u, s) \tag{3}$$

For the scenario presented in Fig. 1, it means, for example, that an utterance *red* allows the literal listener $L_0$ to assign equal probability to the objects in the middle and on the right (a red cloud and a red circle).

Finally, pragmatic interpretation is formalized as a pragmatic listener who uses Bayes rule to solve the inverse problem of recovering the latent state $s$ based on their prior beliefs and the speaker's policy:

$$P_{L_2}(s \mid u) \propto P_{S_1}(u \mid s) \cdot P(s) \tag{4}$$

Unlike the literal listener, who assigns equal probability of being chosen to all objects that qualify, the pragmatic listener $L_1$ imagines a cooperative speaker and infers the intended meaning by reasoning about her communicative behavior. For example, upon hearing *red*, the $L_1$ would reason that if the speaker wanted to refer to the red circle she could have said *circle* since that utterance identifies it uniquely. Since she didn't say *circle*, it must be that she meant the red cloud. The $L_1$ would then assign a higher probability for choosing the red cloud upon hearing *red*.

## RSA models for social meaning

The speaker's behavior in the vanilla RSA model is driven by the goal of signaling the intended meaning efficiently. As such, the vanilla RSA does not cover situations where speakers choose indirect utterances for social reasons, such as politeness. In order to account for such additional social considerations, the utility function of the speaker can be extended to also include additional goals beyond being informative. For example, work on politeness has suggested representing the speaker's utility function as a linear combination of the desire to be informative and to maximize the emotional well-being of the listener [22,23,40].

$$U(u, s, \gamma) = \gamma\ U_{\text{informative}}(u, s) + (1 - \gamma)\ U_{\text{value}}(u, s) \tag{5}$$

Here, the utility component to be informative is as defined above, and the novel social utility component can be defined in terms of how much the literal interpretation of utterance $u$ pleases the listener when the true state is $s$. Politeness models commonly consider scenarios where the speaker needs to give feedback to the performance of the listener, such as in (9).

(9) Politeness scenario:

 a. Alex: How was my poem?

 b. Bo: It was not bad!

Politeness models [22,23,40] predict that even if the the utterance "not bad" is not optimal from the informativity point of view because its meaning is compatible with a wide range of possible world states, it is an optimal choice if the speaker is simultaneously trying to signal

a true state and make the listener happy. Politeness models further predict that if the speaker optimizes solely the social utility ($\gamma = 0$), she would be expected to select only positive utterances. Setting priority to sending fully true information ($\gamma = 1$) would result in a preference for direct utterances. A combination of these goals ($0 < \gamma < 1$) leads polite utterances being chosen, which tend to be indirect.

However, since social utility in previous work on politeness is defined via the emotional value that an utterance carries for the listener, these models do not generalize to other cases of indirectness, where the speaker's goals may not be about producing positive feedback. For example, an utterance, such as (10a) or (10b), where the speaker expresses an opinion about the recent election, does not have a straightforward low or high politeness utility. As a result, politeness models predict that the utterances (10a) and (10b) carry the same social utility. Hence, the speaker's choice in this case will be handled purely by information utility. If the speaker's goal is to communicate a strongly positive opinion (five hearts out of five, if mapped to a simple hearts scale), informational utility will be higher for the direct utterance (10a) compared to the indirect one (10b). The predicate "amazing" in (10a) is only compatible with a state of the world where the speaker would assign five hearts out of five to the election outcome. While "interesting" is also compatible with this world state, "interesting" can also refer to other world states that include fewer hearts. Hence, to express a strongly positive opinion (five hearts), the predicate "amazing" is more useful since it produces a higher chance of the listener inferring the intended state correctly.

(10)   a.   The election outcome is amazing!
         b.   The election outcome is interesting!

Politeness models further predict that a positive evaluation, such as in (10a) has a higher utility than the negative one, such as in (11). However, this prediction fails to capture the fact that a positive evaluation can be dispreferred if the expected listener's opinion is actually negative. Politeness models therefore cannot account for the fact that a negative evaluation, such as in (11) can be an optimal speaker's choice if her true opinion is negative and the expected listener's opinion is negative as well.

(11)   The election outcome is awful!

In the following section we introduce AMIC, which operationalizes the social utility of utterances by formalizing belief alignment objectives. We start with proposing a more general formalization of opinions, which are then used in the alignment model.

## Opinions and their degree of alignment

The general idea of the AMIC is that speakers choose more indirect expressions if they are not sure that their own opinion aligns well with that of the interlocutor(s). Spelling out this idea formally requires making assumptions about how to represent opinions and how to measure alignment between them. It is common in models of opinion dynamics (e.g. [69–71]) to focus on the simplest case of opinions, namely opinions about a binary issue (such as whether abortion should be legal, veganism is good, climate change is human-made, etc.), and to represent an agent's opinion simply as a number $o \in [0; 1]$ on the unit interval. The opinion $o$ is then a single number representing the agent's *position*, i.e., how much the agent agrees with the binary issue. For our purposes, this representation of opinions is not fine-grained enough, because we would like to represent two relevant dimensions:

(i)   **position**: to what extent does the agent tend to agree with the issue?
(ii)  **opinionatedness**: how large or small is the range of positions on the issue that the agent would find acceptable?

We therefore represent an agent's **opinion state** in terms of a probability distribution on the unit interval. As a choice of convenience, we consider the class of Beta distributions (Figure 2). A Beta distribution is usually defined with parameters $\alpha$ and $\beta$. We will use the symbols $\beta_1$ and $\beta_2$, correspondingly to refer to these parameters to avoid confusion with the $\alpha$ parameter of the RSA models. However, for convenience, a Beta distribution can be re-parameterized in terms of its mean $\mu \in [0; 1]$ and "sample size" $\nu > 0$. Starting from $\beta_1, \beta_2 \geq 1$, as the usual parameters of the Beta distribution, this alternative parameterization is obtained via the one-to-one mapping: $\mu = \frac{\beta_1}{\beta_1 + \beta_2}$ and $\nu = \beta_1 + \beta_2 - 2$. The mean $\mu$ can be interpreted as the agent's position or bias, and the sample size $\nu$ can be interpreted as the agent's opinionatedness, where $\nu = 0$ corresponds to a uniform distribution over $[0; 1]$, that is, effectively no, or a fully ambivalent, opinion. As a result, $\nu$ reflects how much evidence the agent has accrued to back up her position. The set of all opinion states $\mathcal{O}$ is then given by all Beta distributions (with $\mu \in [0; 1]$ and $\nu \geq 0$). We denote the listener's and speaker's opinion as $O_L$ and $O_S$ respectively. Fig 2 illustrates five opinions encoded as Beta distributions. To illustrate the effect of parameter combinations on the shape of the Beta distribution further, we have designed a dynamic web-application that is available at https://cognitivemodeling.shinyapps. io/indirectness/ (please check the option "add position-opinionatedness curve" on the right to manipulate the parameter values.).

When representing an opinion by means of a density that is parameterized via two parameters, a measure of alignment between two agents' opinions should be sensitive to both parameters. If we represent opinions as probability distributions, we can use information-theoretic measures of divergence or distance between probability densities, which are sensitive to both expected value and variance of the distributions they relate. Concretely, we want a

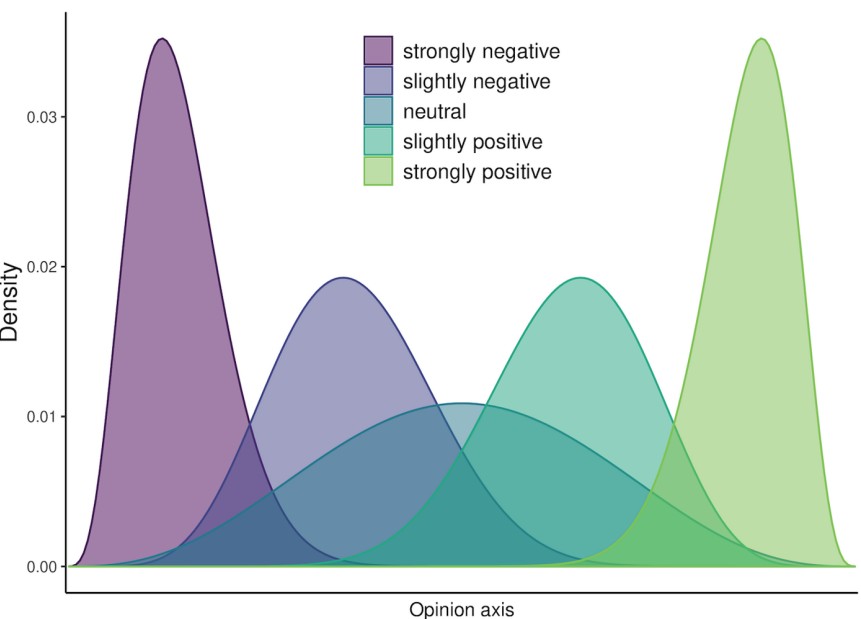

**Fig 2. Examples of different opinion states as Beta distributions with different values for parameters 'position' $\mu$ and "opinionatedness" $\nu$.** The five densities' parameters are (in order of increasing 'position'): $\mu_1 = .14$, $\nu_1 = 35$; $\mu_2 = .36$, $\nu_2 = 22$; $\mu_3 = .5$, $\nu_3 = 8$; $\mu_4 = .64$, $\nu_4 = 22$; $\mu_5 = .86$, $\nu_5 = 35$.

measure of **opinion divergence** to be a function:

$$\text{Div}: \Delta(\mathbb{R}) \times \Delta(\mathbb{R}) \to \mathbb{R},$$

that maps a pair of opinion states onto a real-valued measure of how much the opinion states diverge from each other. As for notation, we write $\Delta(X)$ as the set or space of all probability distributions over $X$. In the following, we use a symmetrized version of Kullback-Leibler divergence to measure alignment. If $P$ and $Q$ are probability distributions, we define opinion divergence as:

$$\text{Div}(P, Q) = D_{\text{KL}}(P||Q) + D_{\text{KL}}(Q||P),$$

where $D_{KL}$ is the KL-divergence. Other information-theoretic measures of divergence or distance are conceivable. In the Supporting information we show divergences between the five opinion states from Fig 2, for symmetrized KL-divergence and some salient alternatives (S1 Fig). Except for the very regular Earth Mover's distance measure, simulations with the alternative divergence measures yield similar qualitative predictions when used in the final model, as illustrated in S2 Fig, S3 Fig, and S4 Fig. The Kullback-Leibler divergence has also been used for information utility calculations in other work within the Rational Speech Act framework, e.g. [10], indicating full compatibility with other utility formalizations. Beyond RSA, KL divergence measures have been employed in the active inference literature modeling combinations of epistemic and goal-directed behavior as well as multi-tasking and task-switching behavior [72–74].

## Higher-order beliefs about opinions

The AMIC assumes that the pragmatic choice of an utterance as well as the interpretation of an utterance are sensitive to opinion alignment. But there can be uncertainty about the interlocutor's opinion (first-order uncertainty), uncertainty about the interlocutor's first-order uncertainty (second-order uncertainty), and so on. While AMIC does not go beyond second-order uncertainty, it may be useful, nonetheless, to have a general notation for potentially even higher-order beliefs.

Let $X$ be the listener $L$ or the speaker $S$, and $Y$ be the respective other agent. If $O_Y$ is agent $Y$'s opinion state, then $\pi_1^X$ is agent $X$'s (first-order) belief about agent $Y$'s opinion. Formally, $\pi_1^X \in \Delta(\mathcal{O})$ is a probability distribution over the space of all opinion states (here: the space of Beta distributions). For any $i > 1$, $\pi_i^X$ is agent $X$'s ($i$-th order) belief about agent $Y$'s ($i$–1)-th order belief. For example, a second-order belief of agent $X$ is a probability distribution $\pi_2^X \in \Delta(\Delta(\mathcal{O}))$, i.e., a probability distribution over probability distributions over Beta distributions. In other words, the second-order belief of $X$, that is, $\pi_2^X$, denotes a distribution over potential first-order beliefs of $Y$, that is, $\pi_1^Y$, about the possible opinions of $X$, that is, $\mathcal{O}_X$.

As for notation, we interpret expressions $\pi_i^X$ as random variables and write $P_{X_1}(O_Y \mid \pi_1^X)$ to represent the probability for a particular opinion $O_Y$. For example, we write $P_{S_1}(O_L \mid \pi_1^S)$ to represent a pragmatic speaker's beliefs about the listener's opinions.

## Literal interpretation

As explained above, the vanilla RSA model grounds out pragmatic reasoning in a layer of literal interpretation, often formally represented as a literal listener. The formulation given above in Equation 3 assumes that there is a semantic meaning function $f: s, u \mapsto [0; 1]$ which maps pairs of states and utterances to truth (or "truthiness") values. For our application, we

are interested in what expressions like in (12) below reveal about the opinion of a speaker. In the present context, we sidestep the linguistic question of how such statements are related to information about opinions (in the wide sense that we endorse here). We will simply assume, for the time being, that there is a literal interpretation function $L_0(u) \in \mathcal{O}$, which assigns to each utterance $u$ a distribution over positions (numbers in $[0; 1]$) usually associated with $u$ based on its conventional meaning. For the simulations reported in this paper, we will provide an empirical measure of $L_0(u)$, as detailed in the description of Experiment 1 below.

## Pragmatic speaker

The AMIC formalizes the situation in which a pragmatic speaker chooses between utterances in an attempt to satisfy multiple conversation goals: (i) signal their own opinion and (ii) align with what they believe to be the listener's opinions, given their first order belief $\pi_1^S$. The latter objective models the active avoidance of opinion conflicts, that is, strong opinion mismatches. For example, the AMIC will assign a higher probability to the utterance (12b) than to (12a) when $\pi_1^{S_1}$ is believed to oppose the speaker's opinion $O_{S_1}$ and the speaker has a strongly positive opinion about some matter. On the other hand, if the speaker has a positive opinion and believes that opinions are aligned, then the model will assign a higher probability to the utterance (12a), since this utterance will have a high information utility and social utility.

(12)   [Speaker's opinion in strongly positive.]

 a.   The election outcome was amazing.

 b.   The election outcome was interesting.

Formally, the mental state of the pragmatic speaker is captured by their own opinion $O_{S_1}$ and their beliefs about the opinion of the listener $\pi_1^{S_1}$. Given $O_{S_1}$, $\pi_1^{S_1}$ and the assumed literal listener's interpretation of utterances, that is, $L_0(u)$, we can define the two goals of the speaker as:

1. **informative goal**: $L_0(u)$ should be as close as possible to the speaker's own opinion $O_{S_1}$, and
2. **social goal**: $L_0(u)$ should be as close as possible to the believed listener's opinion, that is, $\pi_1^{S_1}$.

These two goals translate into two utility functions, where the social utility corresponds to an expected utility over potential opinions of the listener:

$$U_{\text{inf}}\left(O_{S_1}, u\right) = -Div\left(O_{S_1}, L_0(u)\right)$$

$$U_{\text{soc}}\left(\pi_1^{S_1}, u\right) = -\int P_{S_1}\left(O_L \mid \pi_1^{S_1}\right) Div\left(O_L, L_0(u)\right) \, \mathrm{d}O_L$$

The model specification assumes that the speaker knows how the listener interprets utterances—it is part of their background beliefs. Further work may evaluate more complex scenarios where the listener's interpretation is not fully transparent to the speaker. A potential solution lies in including lexical uncertainty into the model [75–77].

We use the information-theoretic notion of Kullback-Leibler divergence, as specified above, in the calculation of both information and social utilities. This divergence measure takes into account both the location of the distribution on the negative-positive scale and how peaked the distributions are. Thus, we are able to consider both the polarity of an opinion and the speaker's certainty. The *total utility* $U_{\text{total}}$ is a linear combination of these two, with

parameter $\gamma$ weighing their relative importance:

$$U_{\text{total}}\left(O_{S_1}, \pi_1^{S_1}, u\right) = \gamma\, U_{\text{inf}}\left(O_{S_1}, u\right) + (1 - \gamma)\, U_{\text{soc}}\left(\pi_1^{S_1}, u\right) \tag{6}$$

The speaker's *utterance choice probability*, given their own opinion and a belief about the literal listener's opinion, is the usual soft-max of the total utility:

$$P_{S_1}\left(u \mid O_{S_1}, \pi_1^{S_1}\right) \propto \exp\left(\alpha\, U_{\text{total}}\left(O_{S_1}, \pi_1^{S_1}, u\right)\right) \tag{7}$$

As an example, Fig 3 shows the probabilities predicted by the AMIC for uttering one of the five opinions given particular opinions of speaker and listener—where the opinions are modeled by the Beta densities shown in Fig 2—and given a relative weighting $\gamma = 0.8$ of the informative utility. The AMIC predicts that speakers are more likely to choose an indirect utterance when they expect the listener to have an opposing opinion. The more the opinions are believed to align, the higher is the probability to choose the most direct opinionated statement. S6 Fig shows further examples of simulations for numerical utilities and resulting speaker probabilities.

## Pragmatic listener

The pragmatic listener $L_2$ uses the utterance-generating model of the pragmatic speaker, in concert with Bayes rule, to infer which mental state of the speaker (consisting of an opinion and a belief about the literal listener) could plausibly have led to the observed utterance. Consequently, the pragmatic listener's mental state is a triple $\langle O_{L_2}, \pi_1^{L_2}, \pi_2^{L_2} \rangle$ consisting of: (i) $L_2$'s own opinion $O_{L_2} \in \mathcal{O}$, (ii) $L_2$'s first-order beliefs $\pi_1^{L_2} \in \Delta(\mathcal{O})$ about the speaker's opinion, and (iii) $L_2$'s second-order beliefs $\pi_2^{L_2} \in \Delta(\Delta(\mathcal{O}))$ about the speaker's beliefs about the listener's opinion. The posterior beliefs of the pragmatic listener about the speaker's opinion are

**Fig 3. Utterance choice: model predictions.** The speaker's actual opinion is strongly positive. Her utterance choice depends on her opinion and the belief about the opinion of the listener, as well as the communicative goal. In this simulation, the informational and social goals are weighted at 0.8 and 0.2, respectively; the $\alpha$ parameter is set to 0.18. A higher value of $\alpha$ leads to more deterministic utterance choices that favor the utterance with highest utility. The left panel demonstrates that when the speaker believes that the listener has a conflicting opinion (strongly negative), she prefers less direct utterances (slightly positive or even neutral) although her opinion is actually strongly positive.

inferred by Bayes rule:

$$P_{L_2}\left(O_{S_1} \mid u, \pi_1^{L_2}, \pi_2^{L_2}\right) \propto \int P_{S_1}\left(u \mid O_{S_1}, \pi_1^{S_1}\right) P_{L_2}\left(O_{S_1} \mid \pi_1^{L_2}\right) P_{L_2}\left(\pi_1^{S_1} \mid \pi_2^{L_2}\right) \mathrm{d}\pi_1^{S_1}, \quad (8)$$

which is the marginal distribution over the opinion $O_{S_1}$, marginalizing over the other component that the pragmatic listener is uncertain about, which is the speaker's beliefs about the (literal) listener $\pi_1^{S_1}$.

## Learning about each other: a simulation

The pragmatic speaker protocol defined in Equation (7) describes a general way of choosing utterances in cases where the communication of opinions is important. Likewise, the pragmatic listener interpretation rule in Equation (8) describes a general format of inferring posterior beliefs about the speaker's opinions after hearing an utterance, based on prior beliefs and the assumption that the speaker generates utterances following the protocol in Equation (7). Together, these production and interpretation rules provide a simple model of learning about each other's opinion (see Fig 4). For example, after a first utterance $u_A$, which Alex chooses based on Equation (7), Bo may update her prior beliefs about Alex's opinion using the rule in Equation (8). The posterior beliefs Bo obtains via Equation (8) may then feed into her choice of a subsequent utterance $u_B$, again chosen via Equation (7). Finally, Alex may then interpret the utterance $u_B$ via Equation (8) to learn from how Bo reacted (via $u_B$) to her utterance $u_A$. In this way, the model sketched here shows a path for agents to learn about each other's beliefs. A particularly interesting possibility is that more sophisticated agents may use the sequential nature of this model to choose utterances strategically, based on their potential to reveal beliefs from anticipated follow-up utterances. For instance, Alex may choose a particular $u_A$ also taking into account how much they will learn about Bo's beliefs from the likely reactions $u_A$ may trigger in Bo. Experimental evidence suggests that at least some speakers are capable of using ambiguity strategically to gain information about the interpreter's prior preferences [20,21], but we will not model this behavior here.

To assess the model's predictions we simulate the interaction shown in Fig 4. In particular, we simulate the belief updates given person A chooses an utterance $u_A$ to which person B provides a response $u_B$. The simulation assumes the informational weight $\gamma = 0.8$—yielding a social weight of $1 - \gamma = 0.2$—and a soft-max factor $\alpha = 0.18$ (Eq. 7). Similar values and similar densities yield similar results. We represent opinions by means of the five Beta distributions shown in Fig 2 and beliefs as probability masses over those five opinions. In particular, we commence with uniform prior beliefs over the five opinion densities as first and second order beliefs (i.e, $\pi_1^A, \pi_1^B, \pi_2^A, \pi_2^B$)—all of which we encode as probability masses over the five opinion densities. Moreover, person A and B are assumed to have one particular opinion $O_A$ and $O_B$, respectively. The first utterance $u_A$ leads to two updates. Person B updates her beliefs of the opinion of person A (i.e., $\pi_1^{B|u_A} \leftarrow \pi_1^B$). By the same process, person A updates what she believes that person B now believes about person A herself, seeing that she has revealed aspects about herself via her utterance $u_A$ ($\pi_2^{A|u_A} \leftarrow \pi_2^A$). After the choice of the response $u_B$ by person B, we finally compute the resulting belief that person A will have about person B's opinion (i.e., $\pi_1^{A|u_B} \leftarrow \pi_1^A$). As we implement $\pi_2^A$ identical to $\pi_1^A$, that is, as a probability mass over five considered opinion densities, the integral effectively sums over all potential opinions that person A may have, when considering her utterance and starting from a uniform prior.

Table 1 shows selected results from these simulations. A full simulation of all possible combinations of utterances and responses can be found in the Supporting information section (S7 Fig). We see that when speaker A's utterance $u_A$ is strongly negative and

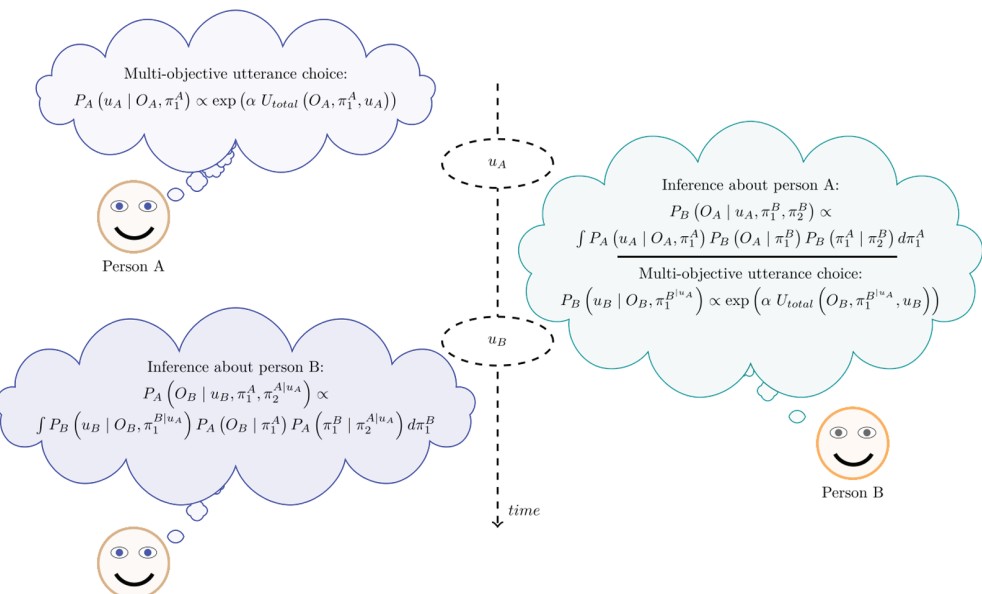

**Fig 4. Multi-turn interaction in the alignment model.** Each agent infers the other agent's beliefs based on their prior beliefs about the interlocutor's beliefs and the utterance probabilities that these prior beliefs about the interlocutor entail. In this plot, we diverge from talking about speakers and listeners and instead talk about persons $A$ and $B$ (e.g., Alex and Bo). This simplifies the notation so that, for example, $\pi_2^B$ are person $B$'s second order beliefs. We denote with $\pi_1^{B|u_A}$ person $B$'s first-order beliefs (about person $A$'s opinions) after interpreting utterance $u_A$ and, similarly, $\pi_2^{A|u_A}$ person $A$'s second-order beliefs (about person $B$'s beliefs about person $A$'s opinion) after having uttered $u_A$.

**Table 1. Predicted probability distributions over inferred speaker B's opinions (i.e., $\pi_1^A$) given a strongly positive, neutral, or negative statement of speaker A and either a slightly negative (*rather bad*) or a slightly positive (*decent*) response of speaker B. Utterances are marked *in italics*, labels of opinions appear in normal font.**

| | A's posterior beliefs about B's opinion | | | | |
|---|---|---|---|---|---|
| | **Strongly negative** | **Slightly negative** | **Neutral** | **Slightly positive** | **Strongly positive** |
| | A: *The election results are terrible* (strongly negative) | | | | |
| B: *I find them rather bad* | 0.16 | **0.35** | 0.28 | 0.19 | 0.02 |
| B: *I find them decent* | 0 | 0.08 | 0.18 | 0.32 | **0.42** |
| | A: *The election results are okay* (neutral) | | | | |
| B: *I find them rather bad* | 0.27 | **0.35** | 0.24 | 0.13 | 0.01 |
| B: *I find them decent* | 0.01 | 0.13 | 0.24 | **0.35** | 0.27 |
| | A: *The election results are amazing* (strongly positive) | | | | |
| B: *I find them rather bad* | **0.42** | 0.32 | 0.18 | 0.08 | 0 |
| B: *I find them decent* | 0.02 | 0.19 | 0.28 | **0.35** | 0.16 |

speaker B chooses a slightly positive response, the model infers that speaker B's actual opinion is most likely strongly positive (second row). A slightly negative response in this situation suggests that speaker B's opinion might be slightly negative (35%) but also neutral (28%), or strongly negative (16%) (first row). If speaker A chooses a strongly positive utterance while speaker B responds with a slightly negative utterance, the model infers that the listener's actual belief is most likely strongly negative (42% chance, previous to last row).

## Summary: Modeling

In this section, we have presented the Alignment Model of Indirect Communication (AMIC), which formalizes the intuition that one reason for the use of indirect utterances is to avoid conflict between revealed opinions. It offers a mechanism based on formalizations of Bayesian inference that allows for the detection of opinion divergences without making them explicit. To represent the meaning of utterances and the opinions of conversation partners, we have used Beta distributions. We have then formalized beliefs about the beliefs of the conversation partner. While the formalization enables an infinite regress, our model uses the recursion up to the second-order belief only, which is probably the typical cognitive limit in everyday conversations. Experimental evidence for postulating cognitive limits comes from the field of decision making and behavioral economics where reasoning about mental states is studied in the context of strategic games. Thus, employing higher-order mental models [78] is associated with higher cognitive effort [79–81]. In communication, evidence for limits on the levels of higher-order representations comes from the study of reference in dialogue [82]. Work on the understanding of sentences that involve propositional-attitude verbs further suggests that speakers are limited in their ability to reason about higher-order mental states [83].

We introduced a pragmatic speaker function $S_1$ that regulates the choice of utterances by balancing the informational and social goals. We have formalized the process of inferring the beliefs of a speaker following her utterance in the pragmatic listener function $L_2$. The AMIC predicts that indirect utterances can become an optimal speaker's choice when she is simultaneously pursuing informational and social goals. It further captures the fact that speaker's opinion may differ from the literal meaning of her utterance. The AMIC also makes non-trivial predictions, as shown in Table 1, about opinion inferences in two-turn dialogues (of the kind shown in Fig 4). In the next section, we report on a set of empirical tests of the model and discuss to which extent AMIC reflects the qualitative patterns we witness in the data.

## Behavioral data

In this section, we report the results of three experiments that were designed to obtain distributions that represent the values of predicates on a five-heart scale (Experiment 1) and test the predictions generated by the alignment model. In particular, Experiment 2 targets the pragmatic speaker behavior, and Experiment 3 assesses how participants draw inferences about opinions, testing the predictions of the pragmatic listener layer of the model.

Data reported in this paper were collected in the period from March 16, 2023 to May 24, 2024. We have obtained written consent from all participants and reimbursed them for their participation at an hourly rate of £10. The Committee for Ethics in Psychological Research of the University of Tübingen has reviewed the experimental protocol for this and other experiments reported in this paper on March 03, 2021 and has not identified any ethical concerns with the protocol or data handling.

### Experiment 1: Empirical baseline of utterance values

In Experiment 1, we obtain an empirical baseline of utterance values mapped to a one-dimensional opinion-space, as it is represented within the literal listener layer of the model. The usual role of the literal listener in RSA models is to anchor pragmatic reasoning in literal interpretation. The AMIC requires a literal listener that captures how various utterances relate to opinion states. Concretely, we consider a literal listener as a function $f$ that maps an utterance into opinion space, so that $L_0(u) \in \mathcal{O}$, where the precise distribution may be determined from empirical data as described next.

To represent the values of utterances on the opinion-space in the form of a distribution, we conducted an online experiment via the Prolific crowd-sourcing platform with US-English speakers($n$ = 51, 27 female, 23 male, 1 other; age range: 19–74, mean = 39, median = 35).

We followed the data-elicitation paradigm proposed in [40], who asked participants to evaluate predicates of personal taste, such as "good" and "not bad", by mapping them to a Likert-scale: the participants assigned a different number of hearts depending on their perception of the description and the stated speaker's goals. In our experiment, we asked the participants to evaluate similar statements within a carrier phrase, like in (13), on a heart-scale from 1 "strongly negative" to 5 "strongly positive":

(13)    The election outcome is...

A sample trial is shown in Figure 5.

We evaluated a total of ten different topics, such as election outcomes, animal condition in agriculture, immigration laws, law enforcement, women's access to reproductive care, etc. As conversation topics we selected those that could potentially elicit a variety of opinions from strongly positive to strongly negative.

To create experimental stimuli, we paired topic frames with one of ten subjective predicates (14). We selected 10 subjective predicates that cover a range from strongly negative, as in (14a), to strongly positive ones, as in (14e), taking inspiration from the choices adopted in politeness studies [23]. Experiment 1 was designed to empirically assess our intuition about the placements of predicates on this scale. Each participant judged 50 sentences that were presented in random order. We therefore collected on average 255 data points per predicate. A sufficiently large number of observations allowed us to abstract away from individual variation in the interpretation of utterances.

(14)    a.  terrible, awful

b.  poor, rather bad

c.  interesting, okay

d.  decent, pretty good

e.  amazing, great

Figure 6 displays the values assigned by the participants to each of these adjectives with all topics pooled together. It is these empirical distributions that we use as a representation

**The city's climate policies are interesting.**

The speaker's attitude is:

Strongly Negative    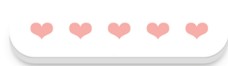    Strongly Positive

Click 'continue' to move on.

Continue

**Fig 5. Experiment 1 (sample trial).** Participants are asked to indicate the speaker's attitude in hearts for ten subjective predicates.

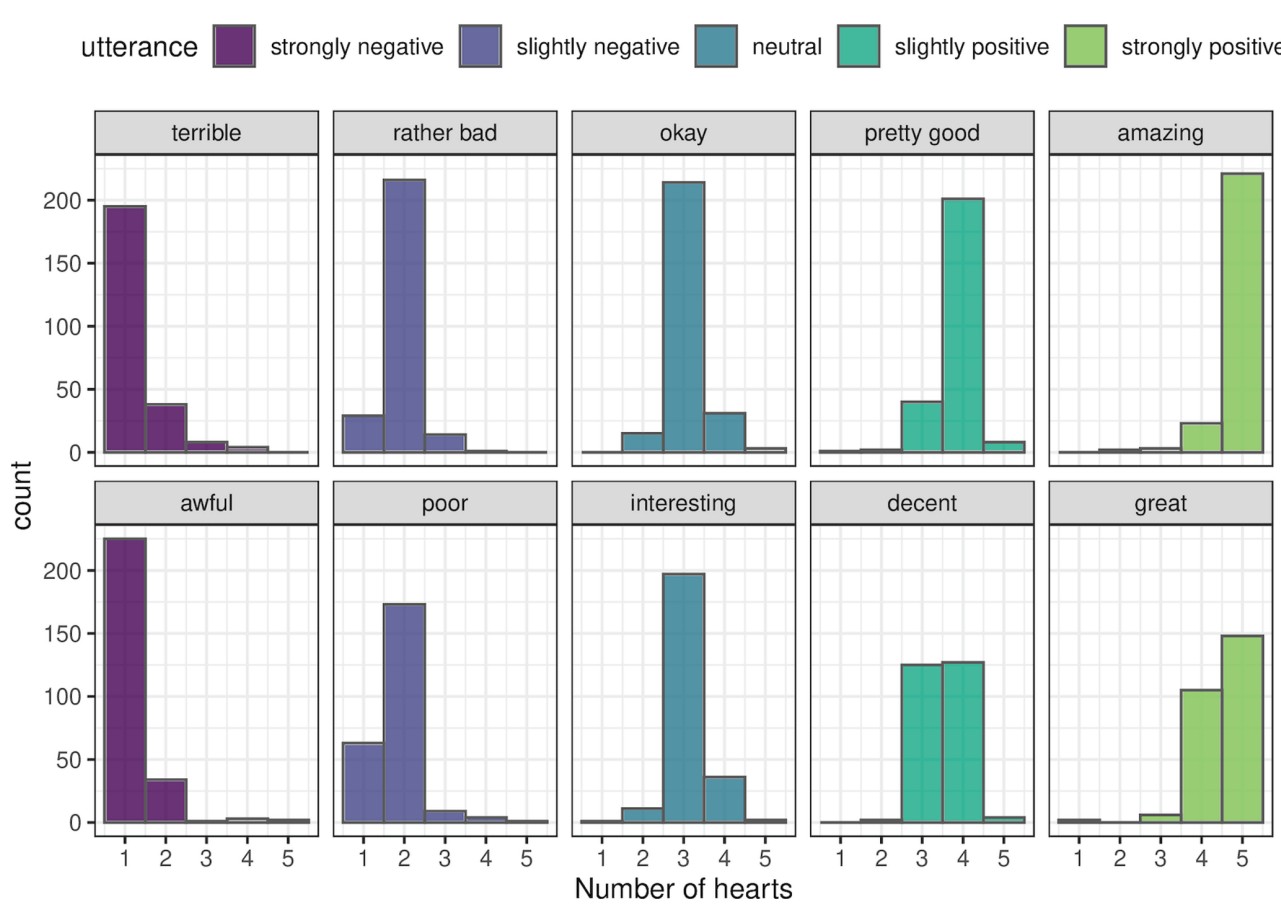

**Fig 6. Utterance values for ten subjective predicates.**

of utterance values. The mean number of hearts assigned to each utterance allows us to classify the utterances as strongly negative (rounded *mean* = 1 heart), slightly negative (2), neutral (3), slightly positive (4), and strongly positive (5). These distinctions are color-coded in Fig 6. Thus, for example, the utterances "terrible" and "awful" are strongly negative, while "amazing" and "great" are strongly positive.

The values we obtained do not directly indicate whether utterances are direct or indirect, since we define indirectness as a property of utterances that emerges in discourse rather than a lexical property of subjective predicates, such as, for example, vagueness. Thus, an utterance, such as (15) may be judged as indirect if the speaker actually has a negative opinion about the election outcome (one or two hearts on our scale). The same utterance can be direct if the true belief state corresponds to four hearts.

(15)   I found the election outcome decent.

In sum, Experiment 1 provides a motivation for assigning the utterances to a scale from "strongly negative" to "strongly positive" and establishes a mapping between these categories and belief states represented in hearts.

## Experiment 2: Pragmatic speaker

Experiment 2 was designed to assess how the communicative goal, the actual opinion of the speaker, and an assumption about the listener's belief affect utterance choices. We collected data from 98 US-English-speaking participants recruited via Prolific (45 female, 52 male, 1 person preferred not to report gender; age range 18–76, mean = 40, median = 35). Data from seven participants were excluded since they reported that they did not fully understand the instructions. Thus, data from 91 participants were entered into the analysis. Fig 7 shows the experimental set up. The experiment was a $2 \times 3 \times 2$ within-subjects design resulting in 12 possible design cells. Since each participant contributed 10 data points, we obtained approximately 76 data points per design cell.

We manipulated the factor "match/mismatch" of whether opinions of speaker and listener matched. The second factor—communicative goals—had three levels: share opinion (informational), share opinion and avoid conflict (informational + social), or simply avoid conflict (social). By informational goals in this context we refer to signaling the true opinion of the speaker. Speakers who pursue only informational goals are expected to always choose utterances that signal their true opinion in the best possible way (e.g., saying that the election outcome is terrible when their opinion corresponds to one heart). Social goals, in turn, amount to avoiding conflict. We expect speakers who pursue only social goals to choose utterances that match the opinion of the listener ignoring their own opinion (e.g., saying that the election outcome is amazing if the listener's state is five heart independent of what the speaker's own opinion is).

**Destiny wants to discuss the election results with Abigail.**

**Here is how Destiny feels about the issue:**

Strongly Negative ❤️ 🤍 🤍 🤍 🤍 Strongly Positive

**Destiny thinks this is how Abigail feels about it, but she is not sure:**

Strongly Negative ❤️ ❤️ ❤️ ❤️ ❤️ Strongly Positive

Destiny wants to share her opinion and wants to be honest about it.

What would <u>Destiny</u> say?

○ The election results are terrible.
○ The election results are poor.
○ The election results are interesting.
○ The election results are decent.
○ The election results are amazing.

Click 'continue' to move on.

[ Continue ]

**Fig 7. Experiment 2 (sample trial).** Participants are asked to select an utterance given the speaker's and listener's opinions and a communicative goal. In this case, the goal is purely informational.

Finally, we also varied whether the speaker's opinion was positive or negative. Concretely, speaker's and listener's opinions were either strongly negative (one heart) or strongly positive (five hearts). These two polar opinions potentially lead to greater differences in utterance choice when we manipulate communicative goals. If the speaker's opinion were neutral, any utterance would not result in a conflict operationalized as divergence between opinions. Therefore to maximize divergence we have selected strongly negative and strongly positive opinions of conversation partners. Four combinations of speaker's and listener's opinions (i.e., strongly positive, strongly positive; strongly positive, strongly negative; strongly negative, strongly positive; strongly negative, strongly negative) allowed us to explore how the presence of conflict (alignment of opinion vs. conflict) affects utterance choice.

In the beginning of the experiment, we informed the participants that a strongly positive opinion corresponds to five hearts, and a strongly negative opinion corresponds to one heart. Each trial contained these labels as well as a reminder. We further offered participants a sample trial before they started the actual experiment.

Based on the association of adjective meanings and the hearts scale established by Experiment 1, the alignment model predicts that speakers should select indirect utterances more often when they anticipate a mismatch in opinions and when they have social goals in addition to informational ones. Concretely, we were interested in two directional hypotheses about the proportions of choices of indirect utterance:

H1: When social goals matter (the informational + social and the social conditions), we expect more indirectness in the mismatch condition than in the corresponding match condition.
   a. mismatch-informational + social > match-informational + social
   b. mismatch-social > match-social
H2: When opinions are mismatched, we expect more indirectness the more social goal matters:
   a. mismatch-social > mismatch-informational + social
   b. mismatch-informational+social > mismatch-informational

Fig 8 shows the proportion of indirect utterances depending on the match/mismatch between the opinions of conversation partners and the speaker's communicative goals. Direct utterances correspond to the speaker's true opinion in hearts. Thus, strongly positive utterances were coded as direct if the true opinion of the speaker corresponded to five hearts. If the utterance matched the polarity of the opinion (e.g. positive), but did not match the degree (slightly positive while the opinion was strongly positive), we counted such utterances as indirect. The "indirect" category further included the neutral utterances. Finally, utterances that did not match the polarity of the true opinion were assigned to the "opposite" category, they constituted 5 % of the trials. The Fig 8 thus demonstrates the proportion of indirect utterances broken by the type of communicative goals and the opinion match/mismatch.

We then focused on the cases of mismatch between the opinions of conversation partners and once again visualized the proportion of indirect utterances. Fig 9 shows how the rate of indirect utterances changes depending on the communicative goal of the speaker and the polarity of the speaker's opinion (strongly negative vs. strongly positive). Alternative figures with non-aggregated data are displayed in the Supporting information section (Figures S8 Fig and S9 Fig).

To test our hypotheses, we ran a single Bayesian logistic regression model in which the dependent variable was binary ("indirect utterance" vs. "other") using the BRMS package in R [84]. The independent fixed effects where all main factors of the experimental design with

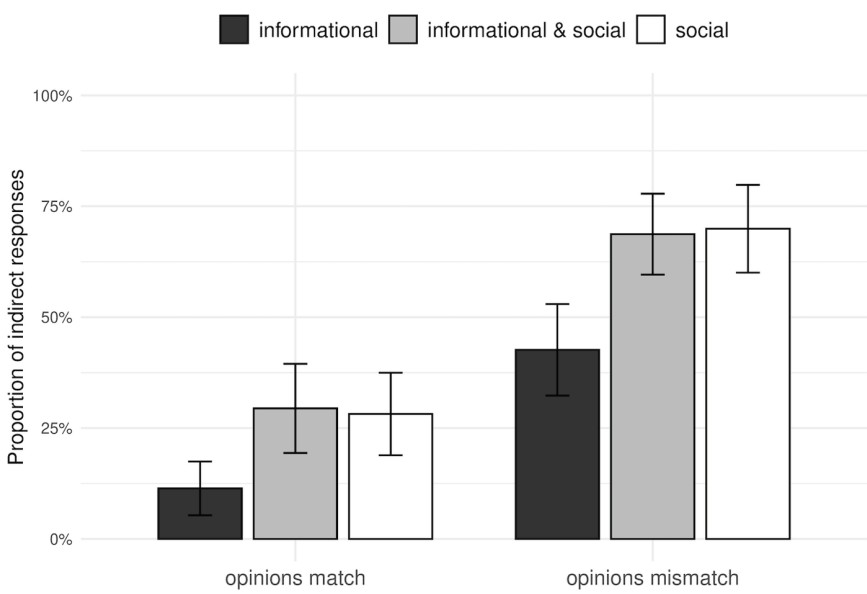

**Fig 8. Experiment 2: Effect of opinions match/mismatch and conversational goals on utterance choice.** Proportion of indirect utterances increases when speakers pursue social goals in addition to informational ones. Participants prefer indirect utterances when the opinions of conversation partners do not match. Error bars represent 95% confidence intervals.

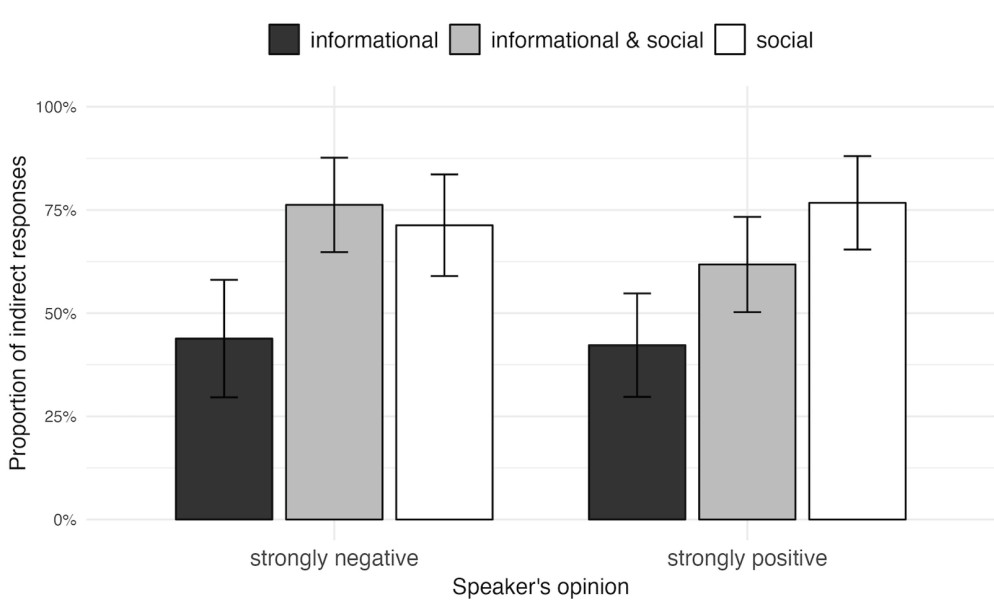

**Fig 9. Experiment 2: Utterance choice when opinions mismatch broken by the polarity of speaker's opinion.** Speakers prefer indirect utterances when they pursue social goals, either alone or in addition to informational ones. This effect holds both for the strongly negative and strongly positive opinions of the speaker. Error bars represent 95% confidence intervals.

all two- and three-way interactions. Additionally, we included by-subject random intercepts. We retrieved samples for the posterior estimates of the predictors of central tendency for each design cell. Posterior samples were aggregated over the relevant subsets of cells and subtracted

to test each one of the four hypothesized contrasts. We take the data and model to provide evidence in favor of a directed contrast if (the sampling-based approximation of) the posterior probability of the difference (aggregated central tendency of higher cell group minus that of the lower) is credibly bigger than zero, which we take to be the case if the 95% credible interval of the difference is entirely bigger than zero.

Based on this analysis protocol, we find that participants chose an indirect utterance reliably more often when the opinions of conversation partners did not match given that the speaker pursued a combination of informational and social goals ($H_{1a}$, posterior mean = 0.412, 95% credible interval = [0.011,0.682]) or a social goal alone ($H_{1b}$, posterior mean = 0.460, 95% credible interval = [0.014,0.736]). Contrary to our predictions, the rate of indirect utterances was not reliably larger for social goals compared to a combination of social and informational goals ($H_{2a}$, posterior mean = 0.0171, 95% credible interval = [−0.147,0.190]). This result suggests that speakers were not able to completely ignore the Gricean maxim of quantity that prescribes speakers to prefer utterances that encode information most efficiently; the social goals condition required them to abandon this principle of cooperative communication. However the comparison of a combination of social and informational goals to purely informational ones conformed to our expectation: speakers were more likely to choose indirect utterances when social goals were on the table in addition to the informational ones ($H_{2b}$, posterior mean = 0.278, 95% credible interval = [0.0005,0.539]).

## Experiment 3: Pragmatic listener

In order to evaluate the model's opinion inference we designed an experiment where the conversation partners exchange opinion statements on a certain topic, and the task of the participants is to infer their actual opinion. The computational model of belief inference presented in Section (12) predicts that the same utterance of the second speaker can be interpreted differently depending on the first speaker's statement and the communicative goals that the participants pursue in the conversation. To mimic the model setup, we informed the participants that the speakers want to exchange opinions but do not want to run into a conflict. We selected six adjectives (out of ten tested in Experiment 1) for the first speaker's utterance such that they reflect a full range of the scale from strongly negative to strongly positive with two adjectives representing the middle of the scale. The second speaker's adjectives included six possible responses and excluded the most opinionated replies (strongly positive and strongly negative), since they were not compatible with the stated communicative goal. Fig 10 displays a sample trial for Experiment 2. Reply of the second speaker appears in the form of "I find…" statements. Previous work in formal semantics has shown that "find" constructions license subjective predicates [54,55].

We collected data from 286 US-English-speaking participants on Prolific (110 female, 173 male, 2 other, 1 chose not to report gender; age range: 19–76, mean = 40, median = 36). Each participant completed 6 trials, each featuring a separate topic. Data from 12 participants were excluded from the analysis since they reported that they did not fully understand the instructions, data from the remaining 274 participants were included into the analysis.

We manipulated the first speaker's statement (from strongly negative to strongly positive) and the second speaker's response (from slightly negative to slightly positive). We limited the second speaker's statement to the range between slightly negative to slightly positive since including strongly negative and positive answers would not match the announced conversation goal that speakers want to exchange opinions but do not want to run into a conflict. We therefore included 32 possible combinations of first and second speaker's utterances, further constraining them in a way that the same adjective cannot appear twice in a dialogue. Since

Cody and Austin meet outside of a club for the first time.
**They would like to exchange opinions but don't want to run into a conflict.**

**Cody says:** The city's climate policy is poor.

**Austin replies:** I find it interesting.

How may <u>Austin</u> actually feel about the issue?

Strongly Negative ♥ ♥ ♥ ♥ ♥ Strongly Positive

Click 'continue' to move on.

Continue

**Fig 10. Experiment 3 (sample trial).** Participants are asked to read a dialogue and indicate how the second speaker actually feels about the issue.

each of the 274 participants contributed six data points, we ended up with approximately 51 data points per design cell.

In critical trials, we then asked how the second speaker may have felt about the topic. In control trials (one trial out of six), we asked the participants to evaluate the statement of the first speaker. This manipulation served two purposes: first, it acted as a way to increase the participant's engagement in the task. And second, the values on the heart-scale associated with the first speaker's utterance provided a baseline that allowed us to order the adjectives on the negative-positive scale and provide an additional confirmation of the scale we obtained in Experiment 1.

Figure 11 shows average values on the heart-scale, alongside model predictions, for opinion inferences based on the first and the second speaker's utterances. A plot of the non-averaged data can be found in the Supporting information section (S10 Fig).

The key qualitative prediction of the model that we would like to assess is one of monotonicity, so to speak: the higher the rank (i.e., the position expressed by the first speaker), the lower the inferred opinion of the second speaker. Thus, for example, the model predicts that participants should assign a higher score to the adjective 'pretty good' if the first speaker statement was negative than when the first statement was strongly positive.

Based on visual inspection, this prediction seems to be supported, at least in tendency, by the data. To test this, we ran a Bayesian regression model, using a cumulative-logit link function to regress the Likert-scale values data against monotonically ordered predictors [85] of the ranks for the first and second speaker's utterances, as well as their interaction, using the default priors of the R package `brms` [84]. We find that the monotonicity coefficient associated with the first speaker's utterance rank is indeed credibly negative (posterior mean: –0.210; 95% credible interval: [–0.314; –0.10]). To further corroborate this result, we also compared this regression model, which has monotonically ordered predictors, against another regression model which allows all rank-levels to be estimated freely from the data (without constraints of monotonic ordering). We find that the model with monotonically ordered factors is substantially preferred under leave-one-out model comparison (difference in expected log-density: 9.1, estimated standard error of this difference: 5.0; see [86]). Taken together, we interpret this as initial evidence in support of the AMIC's predictions.

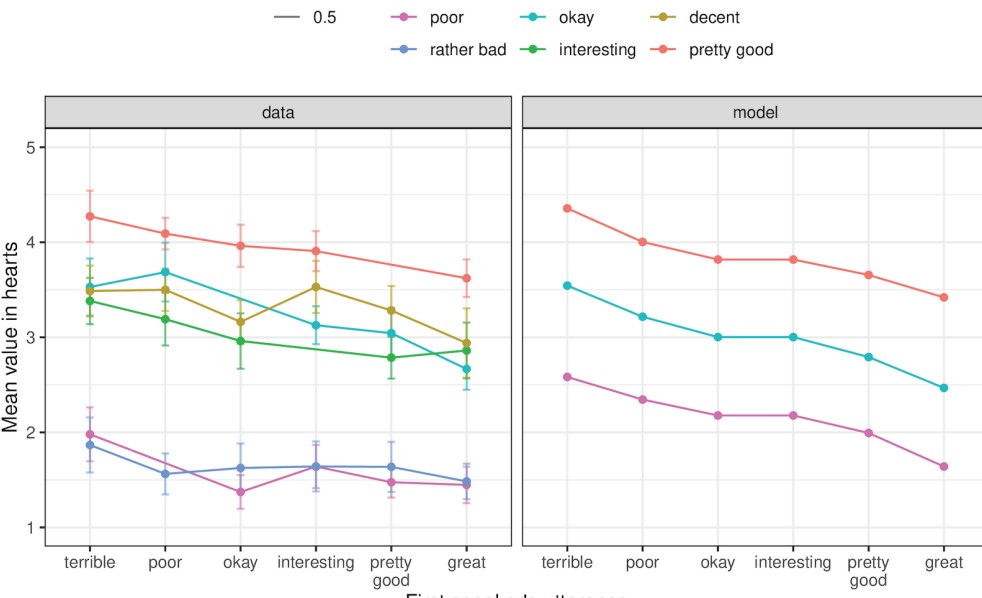

**Fig 11. Opinion inference values.** Panel "data" displays participants' inference of the second speaker's opinion on a five-heart scale (1 heart corresponds to a strongly negative opinion, 5 hearts correspond to a strongly positive opinion). Panel "model" shows corresponding model predictions for slightly positive (*pretty good*), neutral (*okay*), and slightly negative (*poor*) utterances. Error bars represent 95% confidence intervals, model predictions are deterministic. Utilities in the inference model are set to 0.8 (informational) and 0.2 (social).

If we disable social goals in the inference model, then the model interprets the second speaker's utterances at face value independent of the utterance of the first speaker, resulting in lines that are parallel to the x-axis, thus showing no negative slope. A version of Figure 11 with a model that favors only informational goals while social goals are set to 0 can be found in the Supporting information section (S5 Fig.). We further invite the readers to explore how the weighting of informational and social goals affect the inferences in the R Shiny web-app that accompanies the paper. It is available at https://cognitivemodeling.shinyapps.io/indirectness/, tab "Opinion inference".

## Summary: behavioral data

Overall, we have reported the results of three experiments that were designed to provide an empirical assessment of different components of the introduced the AMIC. Experiment 2 targeted the behavior of speakers pursuing a range of communicative goals. The results confirm that social goals and a possible mismatch in the opinions of conversation partners favor the choice of indirect utterances, like the alignment model predicts. Finally, Experiment 3 demonstrates that participants were indeed able to interpret the speaker's responses as indirect when they knew that conversation partners were pursuing social goals. The inferences about the actual speaker B's opinion differed depending on the combination of speakers' contributions. The direction of change corresponds to the one predicted by the alignment model. The empirical findings thus qualitatively support our AMIC model.

## Discussion

One of the goals of theoretical pragmatics is to define how listeners arrive at the meaning of utterances beyond the literal meaning. Game-theoretic models, such as the Iterated Best Response theory [87] and the RSA framework [5,9] have answered this question by assuming that the listener reasons about the speaker, who is, in turn, reasoning about a lower level listener and maximizing the chance of the listener receiving the intended message. Thus, such models defined utterance utility solely by informational utility. Later models included social components, such as politeness [23] and social meaning [18], which additionally influence the speaker's utterance choice. In this paper, we argued that the desire to avoid conflict of beliefs, which we defined in terms of opinion misalignment (or divergence), also affects the types of utterances speakers opt for. We have shown that indirect utterances allow the speaker to simultaneously satisfy informational and social goals. We have furthermore suggested that indirectness is a tool that allows the speaker to probe the state of the listener's beliefs. Thus, conflict avoidance brings the additional benefit of implicitly checking if beliefs are shared.

On top of that, the alignment model of indirect communication (AMIC) proposed in this paper contains a novel inference mechanism that tracks reasoning about mutual opinions over extended stretches of dialogue. The mechanism infers the likely opinion of the listener upon registering her response. The principle behind the implemented inverse inference process is related to inverse reinforcement learning, which is able to infer the reward function that determines the behavior of observed other agents [88]. It is furthermore related to computational models that infer hidden states of other agents, such as their knowledge and their preferences [89,90]—an ability that is indeed already observable in ten month old infants [91]. Our model embeds these mechanistic modeling principles into the realm of conversations, where utterance choices are modeled based on communicative and social objectives. We have used the inverse inference process to update beliefs about the covert opinions of conversation partners. Similar inference processes could be used to, for example, infer conflict avoidance utility weights (parameter $\gamma$ in Eq. 6).

In this paper, we have used computational modeling as a method to formulate our theory of indirect communication. It forced us to be explicit in the choice of representations and processes. It further gave us an opportunity to spell out the contribution of different model components by isolating them in simulations. For example, modeling allowed us to demonstrate how opinion inference in Experiment 3 depends on the combination of speaker's goals. If the model assumes that the second speaker favors informativity and fully ignores social goals, the model bases its opinion inference only on the literal meaning of the utterance. The model fully ignores the first speaker's utterance since the risk of running into conflict is no longer relevant for utility calculation (see the model simulation and its comparison to actual data in the Supporting information section, S5 Fig.) Thus, through simulations we could showcase the contribution of different model components and reveal that both the informational and social utilities are necessary to capture the behavioral patterns we report in Experiments 2 and 3.

Competitor models of politeness [22,23] are unable to match our predictions on the choice of utterances and inference of opinions, since utility components have a simpler structure that does not permit taking into account a potential mismatch of opinions between conversation partners. Social utility of those models is determined solely by the value of an utterance for the listener, with positive utterances carrying a higher value. AMIC makes more fine-grained predictions in two respects: first, it takes into account the opinion of both the speaker and the listener in calculating the utterance utility since it aims to avoid conflict; second, our model predicts that the utterance utility cannot be reduced to a "positive is better"

heuristic since a positive utterance might result in a conflict of opinions if the opinion of the other conversation partner is negative. Furthermore, unlike the existing models of politeness, we can factor in uncertainty over the opinions of the second speaker into our calculation of utterance choice. Politeness models have so far assumed the value of the utterances to be known (e.g. "amazing" corresponds to five hearts, while "terrible" corresponds to one heart). The representation of speaker's opinions as densities allows AMIC to make uncertainties explicit.

AMIC has used Beta distributions as one option to represent uncertainty over a one dimensional opinion axis. Elsewhere, one dimensional magnitude and ordinal axes were proposed to explain, for example, spatial-numerical and spatial-ordinal behavior response effects [92–94]. In the general case, though, opinions may certainly be encoded by means of more complex representations. AMIC could in principle handle such representations, as long as they can be encoded by means of (possibly multidimensional graph-structured) probability densities and information-theoretically updated when additional evidence is perceived. Any such density representation can be used to measure divergences between densities, thus enabling the formalization of AMIC's social and informational utility. As a result, other more complex densities can be implemented in AMIC, as the model is agnostic to the exact method of divergence calculation and Bayesian density updates. Beta distributions can therefore be replaced by other formal densities such as the truncated normal, mixtures of densities, graph-encoded densities, or even density approximations via binning approaches.

We do however, find support for the adequacy of our representational choice in our behavioral data: Beta distributions closely match the empirical distributions of utterance values obtained in Experiment 1. The R Shiny web-app designed for AMIC allows users to explore how/whether the model predictions change when these empirical distributions are used in the calculation of utterance utilities and opinion inference instead of the simulated Beta distributions. The app, along with detailed documentation, is available at https://cognitivemodeling. shinyapps.io/indirectness/. Furthermore, we observe that all qualitative patterns that we reported based on the calculations that rely on Beta distributions remain when these calculations are carried out with empirical distributions from Experiment 1. In sum, while AMIC's functionality is not restricted to Beta distributions as a form of opinion representations, this simplification has enabled to use information-theoretic measures to explain how conversation partners may choose and interpret indirect utterances, extending RSA models to two-turn interactions.

Experiments 2 and 3 further allowed us to verify the predictions of the Pragmatic speaker and Pragmatic listener layers of the model. Admittedly, our experiments are to a certain degree artificial: in Experiment 2, for example, we explicitly asked the participants to follow particular conversational goals, such as "be honest and avoid possible conflict". Conversation goals in actual communication might be significantly more complex than the ones we stated. Moreover, speakers are normally not confined to the narrow choice of utterances that we offered to them. Though restrictive, the controlled experimental environment allowed us to qualitatively assess particular predictions of our model, assuring that the participants and the model have access to the same kind of information—a prerequisite for making model and behavioral data comparisons meaningful.

Allowing participants to have more freedom in their choice of utterances might open new possibilities for investigating the limits of the trade-off between informativeness and conflict avoidance. The utterances that we offered to them so far encouraged finding a compromise between the two. Moreover, we cannot exclude the possibility, that the trade-off becomes less evident at the individual level, as some speakers might follow heuristics in their utterance

choice that encourages them to always/never choose indirect utterances independent of conversation goals. However, at the population level, modeling the relation between informativity and conflict avoidance as a trade-off makes predictions that we were able to attest in experimental settings (for a discussion of individual- vs. population-level modeling in pragmatics see [95].)

Currently, the scope of AMIC is limited to formalizing the inference of the opinion of the speaker given the utterance that she chose. It does not model how the listener may change her own opinion based on that inference—a challenge that we leave for future research. While it may be possible, and ultimately desirable, to derive the way that utterances like "This was interesting!" change an interpreter's opinion just in virtue of their denotational, truth-functional meaning, so far we have concentrated on working out the mechanisms of inferring the speaker's opinion upon observing her utterance. We have treated the semantics of utterances, for the time being, merely in terms of their "opinion-change potential" (OCP), a term coined in intentional analogy to the "context-change potential" of dynamic semantics [96–98,e.g.]. Working out more realistic meaning representations is a challenging but necessary part of possible future modeling efforts. Inclusion of belief alignment as a goal into the utility calculation in AMIC may be considered as a possible extension. Finally, most current models that take social utility into consideration, including ours, focus on a rather narrow set of communication scenarios. Future work should broaden the scope and tackle the development of models that can flexibly adjust their utility calculation to the current communicative context.

Taking a broader view, we propose that indirectness can be viewed as a social means to foster the development of shared opinions, which is possible as long as (i) prior opinions are not fully incompatible from the outset and (ii) the conversation partners are willing to adjust their individual opinions towards those of the conversation partner. From a computational standpoint, a certain degree of flexibility in the belief-encoding distribution ensures that conversation partners are still willing to adjust their belief systems to each other and reach consensus [70]. How exactly listeners update their own opinion based on their prior opinion and on what speakers say will require more elaboration. Factors like trust, status, competence, and the like may play key roles as well as deeper social considerations of utility (do I benefit, in the future, from adopting my neighbors' beliefs?). A simple but compelling algorithm for opinion change is to adapt the parameters of the listener's Beta distribution to be more aligned with the inferred speaker's likely distribution. Taking this path will encourage making contact with models of belief alignment and update, developed within analytic philosophy [99,100], cognitive science [101], and mathematical modeling [102–104].

Conversation analysts confirm that speakers seek contiguity and agreement in verbal communication [42]. Even a mere possibility of facing disagreement on certain topics may lead the speakers to avoid these topics entirely. Thus, a survey of U.S. adults conducted by the PEW Research Center in October 2019 showed that almost half of them stopped discussing politics with other people [105]. A 2022 survey of 66 countries further confirmed this trend [106]. These results show that social considerations and relationship preservation have a substantial influence on speaker's choices, and the preference not to engage in a conversation is one of the extreme but apparently frequent strategies of favoring social utility over the informational one. While allowing conversation partners to avoid conflict, this strategy also carries significant negative consequences for finding alignment and reducing polarization (see [107] for a recent overview). Indirect communication, in this light, opens opportunities for finding a trade-off between various conversation goals that does not lead to the complete abandonment of certain topics. Therefore, understanding the strategies that govern indirect communication gains new meaning in the current age of political polarization.

## Conclusion

In this paper, we have brought together literature from social psychology, philosophy of language, psycholinguistics, and cognitive modeling to formalize the mechanisms that may underlie opinion alignment through the use of indirect utterances. From a sociological perspective, discovering whether opinions are shared serves two purposes: understanding the world through validating reality and belonging to a group [26,108]. The discovery of shared aspects signals to conversation partners that they may belong to the same social group. The discovery of unexpected or rare alignment between two personal characteristics may lead to an even stronger bonding effect [109]. Thus, confirming that certain assumptions belong to the common ground may create the bonding and the "linguistic intimacy" [110] that emerges when an indirect utterance was apparently interpreted as intended.

The proposed model formalizes how production and interpretation of indirect utterances continuously provides conversation partners with signals of whether their belief systems align. While avoiding conflict, they monitor each other's interpretation of indirect utterances and draw inferences about each other's opinions. These inferences open space for dynamic belief alignment and contribute to establishing and maintaining social bonds while speakers navigate complex social environment.

## Supporting information

**S1 Fig. Alternative divergence measures.** Measures of divergence between the opinion distributions may be interpreted as encoding the effort to change one belief into another one, or, in other words, belief compatibility. a) Kullback Leibler (KL) divergence; b) Bidirectional KL-divergence; c) Earth Mover's Distance; d) Jensen-Shannon Divergence.
(PDF)

**S2 Fig. Model predictions with divergence set to unidirectional KL-divergence.** Unidirectional KL-divergence produces a similar qualitative pattern compared to the bidirectional KL-divergence: the model infers a more positive opinion of speaker B given a more negative utterance of speaker A.
(PDF)

**S3 Fig. Model predictions with divergence set to bidirectional Jensen-Shannon Divergence.** Jensen-Shannon Divergence (Fig (15)) shows a weaker relationship between the first speaker's utterance and the inferred opinion.
(PDF)

**S4 Fig. Model predictions with divergence set to Earth Mover's Distance.** The Earth Mover's Distance (Fig (15)) even with modified parameters fails to capture the qualitative pattern observed in the data.
(PDF)

**S5 Fig. Model predictions with divergence set to bidirectional KL-divergence, informational goals set to 1.0, social goals set to 0.** The inferred second speaker's opinion does not depend on the utterance of the first speaker: the model infers that the speaker says what she means, since the social goal has been set to 0.
(PDF)

**S6 Fig. Utterance utility values (panel A) and corresponding utterance choice probabilities (panel B).** The calculations assume that the speaker's opinion corresponds to a strongly positive ($\alpha = 30, \beta = 5$) opinion distribution. Panel A shows utility values for utterances (rows) given different speaker beliefs about the listener's opinion state $\pi_1^{S_1}$ (here assumed to be

single-peaked distributions). Panel B shows the corresponding utterance-choice probabilities computed via Equation 7 assuming $\omega_{inf} = 0.8$, $\omega_{soc} = 0.2$, and $\alpha = 0.18$. The values show that the model generates progressively smaller utilities for utterances that diverge from the speaker's opinion (strongly positive, in this case). Generally, utterances that offer the best compromise between the speaker's opinion and the believed listener's opinion are preferred.
(PDF)

**S7 Fig. Opinion inference.** Model's posterior estimation of speaker B's opinion computed via Equation 8 given an initial utterance that is strongly negative (panel A), strongly positive (panel B), neutral (panel C), or slightly positive (panel D). Each row in each matrix encodes a particular posterior belief distribution $\pi_1^A$ over speaker B's opinion given her response indicated in each row.
(PDF)

**S8 Fig. Experiment 2: Utterance choice (raw data).** The left column shows the cases were the opinions of conversation partners match. Here speakers prefer utterances that correspond to their true opinion. The right column shows the cases of mismatch in opinion. Utterance choices shift towards the middle of the scale.
(PDF)

**S9 Fig. Experiment 2: Utterance choice (raw data), opinions mismatch.** Cases where the opinions of conversation partners do not match. The top row corresponds to the strongly negative opinion of the speaker, the bottom row shows the strongly positive opinion of the speaker.
(PDF)

**S10 Fig. Utterance values for ten considered adjectives.** Each data point represents a participant's response. Jitter was added for visualization purposes. The location of the clusters along the vertical axis reflects how positive the inferred opinion is. Thus, the relevant contrasts lie within each facet between the adjectives at the opposite ends of the scale. The model predicts that upon hearing a predicate, such as "interesting", participants should infer the opinion as more positive if the predicate follows a strongly negative statement compared to a strongly positive statement.
(PDF)

## Acknowledgments

We are grateful for the feedback we received on the earlier drafts of this manuscript from Susanne Winkler, Esme Winter-Froemel, Christian Stegemann-Philipps, Daniel Harris, Elin McCready, Nicole Gotzner, Greg Scontras, Natasha Korotkova, Todd Snider, the audience of XPRAG.it 2024 and the workshop "Background beliefs in the construction of meaning" (Tübingen, January 2025). We would also like to thank our research assistant Antonia Höfer for co-designing and implementing the R Shiny web-app for the model. We acknowledge support from the Open Access Publication Fund of the University of Tübingen.

## Author contributions

**Conceptualization:** Asya Achimova, Michael Franke, Martin V. Butz.

**Data curation:** Asya Achimova, Michael Franke, Martin V. Butz.

**Formal analysis:** Asya Achimova, Michael Franke, Martin V. Butz.

**Funding acquisition:** Asya Achimova, Martin V. Butz.

**Investigation:** Michael Franke, Martin V. Butz.

**Methodology:** Asya Achimova, Michael Franke, Martin V. Butz.

**Supervision:** Martin V. Butz.

**Visualization:** Asya Achimova, Martin V. Butz.

**Writing – original draft:** Asya Achimova, Michael Franke, Martin V. Butz.

**Writing – review & editing:** Asya Achimova, Michael Franke, Martin V. Butz.

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
