## [Decision Letter · Decision Letter 0]

6 Feb 2025

PONE-D-24-57287The alignment model of indirect communicationPLOS ONE

Dear Dr. Achimova,

Thank you for submitting your manuscript to PLOS ONE. After careful consideration, we feel that it has merit but does not fully meet PLOS ONE’s publication criteria as it currently stands. Therefore, we invite you to submit a revised version of the manuscript that addresses the points raised during the review process. While both reviewers expressed positive opinions regarding your work, they also suggested that the manuscript needs significant revision, particularly regarding the methodology and justification of the model. Please respond specifically to these main points when submitting your revised manuscript.

We look forward to receiving your revised manuscript.

Kind regards,

Gareth J. Baxter

Academic Editor

PLOS ONE

Journal Requirements:

Reviewers' comments:

Reviewer's Responses to Questions

**Comments to the Author**

1. Is the manuscript technically sound, and do the data support the conclusions?

Reviewer #1: Yes

Reviewer #2: Yes

2. Has the statistical analysis been performed appropriately and rigorously? 

Reviewer #1: Yes

Reviewer #2: Yes

3. Have the authors made all data underlying the findings in their manuscript fully available?

Reviewer #1: No

Reviewer #2: Yes

4. Is the manuscript presented in an intelligible fashion and written in standard English?

Reviewer #1: Yes

Reviewer #2: Yes

5. Review Comments to the Author

Reviewer #1: This is a very good article that deserves publication, pending some revisions regarding, in particular, the distinction between vagueness/indirectness and the methodological section, where important information is missing.

Specific comments:

Introduction: the authors’ rationale could also be connected with the idea that the meaning of an evaluative statement is assessed against other possible alternatives, as in the case of negative statements (e.g., not overtly bright is assessed against not bright, stupid, etc.), which are also mentioned later on.

L73: You could provide some references to indirectness, which is defined in somewhat vague terms in the current state of the manuscript.

E.g., Ruytenbeek, N. (2024). Indirectness. In Handbook of Pragmatics: 27th Annual Installment (pp. 101-127). John Benjamins Publishing Company. (for an overview of different notions of indirectness and a critical discussion).

L85: it is unclear if this has to do with vagueness or indirect communication; in the philosophy of language, vagueness is a distinct concept, and it is more specific to lexical items than to constructions or utterances.

On vagueness, see e.g., Égré, P., & Klinedinst, N. (2011). Introduction: Vagueness and language use. In Vagueness and language use (pp. 1-21). London: Palgrave Macmillan UK.

L88: on a possible loss of face, it seems to me that this is not only relevant for the speaker but also for the addressee of the comment, who might be offended by do opinion conveyed. So, both the speaker and the addressee would be subject to possible face loss in the situation that you envisage.

L96: regarding the coefficients in the equation, it would help if you could add an example with a specific value for the w.

L176: there is one parenthesis too much: (u, s)).

L204: delete “the pragmatic listener”.

L224: about the politeness scenario, it is unclear to me why you are assuming that pretty good, including a rising intonation (indicated by the exclamation mark, I suppose) would be a case of indirect communication. The fact that this is a positive assessment is not at all ambiguous; therefore, it would not qualify as an indirect speech act.

L243: “Which is higher IN the direct utterance”.

You could also explain more clearly that amazing is compatible with fewer opinion statements compared to interesting, which means that amazing is more informative than interesting. (If I understood this correctly.)

L272-287: it makes sense to conceptualise agreement as a graded concept, but it would be helpful to illustrate the formalization using a concrete, real life, example.

L340: What does “multi-objective” mean?

L398: listener’s �� listeners.

L398-401: this should rather be part of the general discussion or the conclusion of the paper.

L418-425: at some point, you consider the possibility that the goal of communicating an opinion indirectly with a mitigated statement is a way to check speaker-recipient opinion alignment, which could be considered as a speaker-oriented goal. You also explain that you will not include this assumption in your model, but maybe you could add some clarification in the paper about which additional assumptions could (or could not) be incorporated in future elaborations of the model.

In Table 1, it is unclear what corresponds to A’s beliefs and what corresponds to A’s utterances.

L464-5: about the typical cognitive limitation in everyday conversation, please add relevant references.

L484 on ethics: could you add the reference of the application and also how much money the participants received for taking part in the different studies.

For the three experiments reported in the paper, it is necessary to include social demographic information about the sample of the population who participated in the studies, such as age (e.g., median, range) and gender in particular.

I am concerned that no information is provided about the sample size calculation (especially as the number of Ps for each experiment is different), the language of the stimuli, how the stimuli were selected, for instance, why these 10 topics, and whether or not the stimulate were presented in the random order.

L527: word semantics, do you mean “sentence semantics” or “compositional meaning”? As I wrote earlier, it is not totally clear to me that indirectness is a notion that applies to lexical items (unlike vagueness). This is an issue that should be clarified in the manuscript.

L538 and below: data WERE excluded (research data is a plural noun).

L545: How is it possible not to have an informational goal when making an assertion? This should be clarified. I can think of examples, such as stating the obvious (also in indirect requests: “It’s hot in here”, meaning “Please open the window”), but it seems that you have something different in mind, here.

L547: why did you only use very positive and very negative opinions for the experimental stimuli? This decision should be motivated.

The scale on Figure 7 is ambiguous, as five hearts could also mean strongly negative. It will be important to add more information about the instructions that were given to the participants, specifically about how they were supposed to interpret the scale.

L641: It is unclear what “a score” means, in the sense that it is difficult to understand the diagram. In other words more information about the dependent variable should be included.

L701-2: Again, the theoretical background pertaining to the distinction between vagueness and indirectness should be improved to strengthen the interdisciplinary impact of this important piece of research.

Reviewer #2: The manuscript by Dr. Asya Achimova and colleagues proposes an extension of the Rational Speech Act framework by incorporating social value functions to model the computational processes underlying conversational exchanges. The study primarily investigates simplified two-utterance interactions. Although the computational framework is elegantly formulated for this restricted setting, several key issues related to the treatment of communication context and underlying assumptions need to be addressed to bolster the study’s generality and applicability.

Major Points

1. Contextual Articulation of Social Value

The manuscript would benefit from a clearer articulation of the specific communicative contexts in which the proposed formulation of social value is most applicable. For example, in today’s politically polarized climate—where listener opinions can be strongly divided and direct, unambiguous statements might be less favored—it is important to explain the relevance and limitations of the current formulation. A discussion of how these polarized contexts, which may be particularly prevalent in certain Western societies, influence the model’s assumptions would strengthen the justification for the proposed utility function.

2. Dimensionality of Communication Space

The assumption of a one-dimensional communication space in the context of polarized discourse requires further justification. The authors should elaborate on why and how the inherently multidimensional nature of real-world communication can be effectively reduced to a single dimension in this model. Providing theoretical or empirical support for this design choice is essential for establishing the generality of the framework.

3. Treatment of Ambiguity in Indirect Communication

The relationship between indirectness and ambiguity is nuanced. Indirect communication can stem from a compromise between expressing one’s true intentions and preserving the interlocutor’s face, potentially resulting in ambiguity that ranges from a narrow compromise to a multimodal distribution of meanings. The current use of a beta distribution may not fully capture these possibilities. It would be helpful for the authors to test and discuss how their model accounts for various forms of ambiguity inherent in indirect speech.

4. Task Appropriateness for Socially Motivated Indirect Speech

There is a concern regarding whether the task employed in the study adequately captures the complexity of socially motivated indirect speech. For instance, expressions like “interesting” can convey dual meanings in indirect communication. The current experimental design may restrict the expression of such duality. Clarification on how the task allows for or limits the demonstration of these nuanced communicative intentions would be valuable.

5. Informativeness versus Social Pay-off Trade-off

The manuscript posits a trade-off between informativeness and social pay-off. However, it is worth questioning whether this trade-off accurately reflects real-world communicative behavior. A more detailed discussion, potentially supported by empirical evidence, on how this relationship manifests in natural interactions would help substantiate the model’s assumptions.

6. Simulation of Motivational Manipulation

Given that the alignment with the listener’s opinion appears to be a key motivational factor in the model, it is surprising that this motivation is not manipulated within the simulations. While Experiment 2 is consistent with the experimental design, it would be instructive to see how this manipulation might influence outcomes in a subsequent experiment (e.g., Experiment 3). The authors should discuss potential simulation results and their implications for the model’s predictive power.

Minor Points

1. Clarity of Expression in Examples

The example labeled “5-a” appears to be an overly indirect or ambiguous expression of negative opinion. Consider revising this example to include a more direct expression, which would help clarify the intended communicative intent.

2. Quality of Figures

Figure 4 is currently too blurry to be clearly interpretable. Please provide a higher-resolution version to ensure that all details are visible.

3. Comparison with Politeness Models

The manuscript would benefit from a clearer comparison with existing politeness models, especially those previously proposed by the authors. Highlighting the novel contributions of the current study relative to these models will clarify its significance.

4. Data Presentation and Variability

The figures currently lack error bars, which are crucial for conveying the variability of responses across subjects. Please include error bars in the relevant figures to enhance the clarity and reliability of the data presentation.

5. Model Comparison and Alternative Explanations

The data presented appear to be interpretable by several different models. It would be helpful to discuss potential alternative models and to provide a comparison of their explanatory power relative to the proposed model.

6. Figure Axis Legend in Figure 11

The y-axis legend in Figure 11 requires additional clarification. Please provide more detailed information regarding what the y-axis represents to ensure that the figure is self-explanatory.

6. PLOS authors have the option to publish the peer review history of their article (what does this mean?). If published, this will include your full peer review and any attached files.

Reviewer #1: **Yes: **Nicolas Ruytenbeek

Reviewer #2: No

---

## [Author Response · Author response to Decision Letter 1]

4 Apr 2025

Please find the Revisions letter attached.

---

## [Decision Letter · Decision Letter 1]

16 Apr 2025

The alignment model of indirect communication

PONE-D-24-57287R1

Dear Dr. Achimova,

We’re pleased to inform you that your manuscript has been judged scientifically suitable for publication and will be formally accepted for publication once it meets all outstanding technical requirements.

Kind regards,

Gareth J. Baxter

Academic Editor

PLOS ONE

Additional Editor Comments (optional):

While we were not able to receive second reports from both original reviewers, based on the opinion of the first reviewer, and assessing the modifications made and your comments in response to the report of reviewer one, your manuscript clearly meets the criteria for acceptance.

I thank you for your careful consideration of all reviewer comments and your thorough responses.

Reviewers' comments:

Reviewer's Responses to Questions

**Comments to the Author**

1. If the authors have adequately addressed your comments raised in a previous round of review and you feel that this manuscript is now acceptable for publication, you may indicate that here to bypass the “Comments to the Author” section, enter your conflict of interest statement in the “Confidential to Editor” section, and submit your "Accept" recommendation.

Reviewer #1: All comments have been addressed

2. Is the manuscript technically sound, and do the data support the conclusions?

Reviewer #1: Yes

3. Has the statistical analysis been performed appropriately and rigorously? 

Reviewer #1: Yes

4. Have the authors made all data underlying the findings in their manuscript fully available?

Reviewer #1: Yes

5. Is the manuscript presented in an intelligible fashion and written in standard English?

Reviewer #1: Yes

6. Review Comments to the Author

Reviewer #1: The authors have done a great job addressing the comments provided by the two reviewers. I appreciate, in particular, the significant improvements in the literature review and theoretical background sections, where the authors now refer to the key notion of indirectness in speech act theory and are more precise regarding the concept of vagueness and ambiguity. The discussion section too had been expanded, with an acknowledgement of the study limitations and new paragraphs on belief alignment and conversational goals. Finally, I noticed more concrete examples and additional explanations about the formalism used (for less familiar readers).

At this stage, I don't have any more comments or requests for revisions.

7. PLOS authors have the option to publish the peer review history of their article (what does this mean?). If published, this will include your full peer review and any attached files.

Reviewer #1: **Yes: **Nicolas Ruytenbeek

---

## [Editor Report · Acceptance letter]

PONE-D-24-57287R1

PLOS ONE

Dear Dr. Achimova,

I'm pleased to inform you that your manuscript has been deemed suitable for publication in PLOS ONE. Congratulations! Your manuscript is now being handed over to our production team.

Kind regards,

on behalf of

Dr. Gareth J. Baxter

Academic Editor

PLOS ONE